



# Technical Note: On the intercalibration of HIRS channel 12 brightness temperatures following the transition from HIRS 2 to HIRS 3/4 for ice saturation studies

Klaus Gierens[1] and Kostas Eleftheratos[2]

[1]Deutsches Zentrum für Luft- und Raumfahrt, Institut für Physik der Atmosphäre, Oberpfaffenhofen, Germany
[2]Faculty of Geology and Geoenvironment, University of Athens, Athens, Greece

*Correspondence to:* Klaus Gierens (klaus.gierens@dlr.de)

**Abstract.** In the present study we explore the capability of the intercalibrated HIRS brightness temperature data at channel 12 (the HIRS water vapour channel; $T_{12}$) to reproduce ice supersaturation in the upper troposphere during the period 1979–2014. Focus is given on the transition from the HIRS 2 to the HIRS 3 instrument in the year 1999, which involved a shift of the central wavelength in channel 12 from $6.7\,\mu m$ to $6.5\,\mu m$. It is shown that this shift produced a discontinuity in the time series of low

$T_{12}$ values ($< 235\,K$) and associated cases of high upper–tropospheric humidity with respect to ice (UTHi$> 70\%$) in the year 1999 which prevented us from maintaining a continuous, long term time series of ice saturation throughout the whole record (1979–2014). We present that additional corrections are required to the low $T_{12}$ values in order to bring HIRS 3 levels down to HIRS 2 levels. The new corrections are based on the cumulative distribution functions of $T_{12}$ from NOAA 14 and 15 satellites (that is, when the transition from HIRS 2 to HIRS 3 occurred). By applying these corrections to the low $T_{12}$ values we show

that the discontinuity in the time series caused by the transition of HIRS 2 to HIRS 3 is not apparent anymore when it comes to calculate extreme UTHi cases. We come up with a new time series for values found at the low tail of the $T_{12}$ distribution, which can be further exploited for analyses of ice saturation and supersaturation cases. The validity of the new method with respect to typical intercalibration methods such as regression–based methods is presented and discussed.

## 1 Introduction

Ice supersaturation is a frequent phenomenon in cold regions of the troposphere (below $0°C$, in particular in the upper troposphere), important for the weather state, cirrus cloud formation and climate (Gierens et al., 2012). The probability density function of the degree of ice supersaturation is approximately an exponential distribution with a mean supersaturation value of about 15%. A slight change of the mean value implies a large change in the tail of the exponential distribution, thus conditions for in–situ cirrus formation can occur much more frequently or much more seldom than today after a slight change of the mean

supersaturation. Such subtle changes cannot reliably be predicted with climate models; hence the prediction of future cirrus coverage is ticklish. Moreover, cirrus clouds are a component of the climate system and their feedback on climate change is one of the most uncertain issues in climate research (e.g. Ou and Liou, 1995; Stephens, 2005). Any short or long term change in the frequency of occurrence of ice supersaturation and in its probability density function is expected to have an influence on the





cirrus cloud field and therefore on climate change (e.g. Irvine and Shine, 2015). Relatively few papers appear in the literature describing the large and small scale distribution and seasonal, annual and longer time scale changes of relative and absolute humidity of the upper troposphere. A lack of observations, especially those at regional and global scales, has hampered our ability to study the changes in this important climate variable.

An ideal data set to study long–term changes of upper–tropospheric humidity (UTH) is provided by the series of polar or-biting satellites of the National Oceanic and Atmospheric Administration (NOAA), which has started in the late 1970s and is still ongoing, meanwhile in co–operation with the European Organisation for the Exploitation of Meteorological Satellites (Eumetsat). The satellites all carry the High–Resolution Infrared Radiation Sounder (HIRS). Channel 12 of this instrument can be used to retrieve UTH. It is a radiance based quantity that represents a weighted mean over a vertical profile of relative humidity

with a peak of weighting function in the upper troposphere. The retrieval method has been developed by Soden and Bretherton (1993) and improved by Jackson and Bates (2001).

All the NOAA satellites from N06 (launched 1979) to N14 (launched 1994) carried version 2 of the HIRS instrument while from N15 on (launched 1998) version 3 and later version 4 of HIRS was installed. The transition from HIRS 2 to HIRS 3 involved a shift of the central wavelength in channel 12, from $6.7\,\mu m$ to $6.5\,\mu m$. Unfortunately, this is not a small change as it

may appear. The atmosphere is nearly 1.5 times as opaque at $6.5\,\mu m$ than at $6.7\,\mu m$ (see, for instance, the black curve in fig. 1 of Shi and Bates, 2011). Thus the kernel function for the retrieval of UTH peaks about one kilometre higher in the atmosphere for HIRS 3 and 4 than for HIRS 2 (cf. fig. 2 of Gierens and Eleftheratos, 2016), or in other words, channel 12 of N15 and the later satellites is sensitive to a more than $1\,km$ higher layer in the atmosphere than channel 12 of the older satellites of the NOAA series; yet the layers strongly overlap due to large half widths of the corresponding weighting kernels of, say, 4

to 5 kilometres. As temperature decreases on average by $6.5\,K\,km^{-1}$ in the troposphere, the change of the wavelength and the corresponding increase of the weighting function peak altitude led to a discontinous shift in the corresponding brightness temperatures of about $8\,K$ (Shi and Bates, 2011; Chung et al., 2016).

Such a strong discontinuity would break the desired long–term time series, but Shi and Bates (2011) were successful in solving the problem. They performed an intercalibration of the channel 12 brightness temperature, $T_{12}$, of all NOAA satellites,

using N12 as a reference. They compute for each satellite monthly and zonal averages, with $10°$ latitude belts centred on $85°$ S to $85°$ N. Thus, they obtain a set of mean brightness temperature values $T_{L,YM}^N$, where the upper index $N$ is satellite number, and the lower indices are latitude belt and year/month combination. Biases are then computed as individual differences $T_{L,YM}^N - T_{L,YM}^{N+1}$, that is for pairs of subsequent satellites operating in the same months and years. The individual bias values are then put into $5\,K$ wide classes of brightness temperatures. The result of this is a data set providing temperature dependent

corrections for each satellite pair. These corrections are applied pixel–wise (i.e. not simply by adjustment of the time–series means), with N12 taken as reference. The intercalibration procedure solves not only the problem with the wavelength change, minor changes due to variations in filter functions and calibration loads are covered automatically as well.

The inter–calibrated HIRS brightness temperature (BT) data for the past 35 years (1979–2014) have been used to study long term changes in the upper tropospheric water vapour (Chung et al., 2016). With this long term data set we can also study the

upper tropospheric humidity with respect to ice (UTHi, Gierens et al., 2014).



In the present paper this radiance based quantity is used for the first time to study ice supersaturation cases in the upper troposphere with such a long time series. As ice-supersaturated layers are typically much shallower than the layer where channel 12 of HIRS is sensitive to, only a very small fraction of UTHi values exceeds 100% (Gierens et al., 2004). Yet one can argue that there is sometimes ice supersaturation in the upper troposphere when UTHi is of the order 70% and that the

5 probability of occurrence of ice supersaturation increases with the measured value of UTHi in a certain fashion (Lamquin et al., 2009; Dickson et al., 2010). The research focuses on UTHi values exceeding 70% and higher thresholds. Preliminary findings show that the extreme UTHi situations might have increased in the past decade, whereas the zonal mean UTHi remained almost unchanged. These results are very interesting; they contribute to an ongoing debate whether the free troposphere is moistened as a consequence of global warming (e.g. Paltridge et al., 2009; Dessler and Davis, 2010).

Chung et al. (2016) stated that the discontinuity in the time series caused by the transition of HIRS 2 to HIRS 3 has been almost completely removed by the calibration process conducted by Shi and Bates (2011), in which the influence of the filter change was adequately taken into account by a scene radiance–dependent bias correction. Indeed, there is no evidence for a discontinuity in their time series of $T_{12}$ anomalies in the period 1979 to 2015. Although this is true for the mean $T_{12}$, two interesting questions raised here are a) whether Shi and Bates's inter–calibration process is also valid for values found at the

low tail of the distribution of $T_{12}$ when it comes to calculate extreme UTHi cases as in our case, and b) whether it is actually correct to combine the two HIRS time series (HIRS 2; 1979–2005 and HIRS 3/4; 1999–2014) into a single one for the case of low $T_{12}$ values, given that HIRS 2 and HIRS 3 actually sense different layers in the upper troposphere. Assuming that we can physically combine the two time series into one, like Chung et al. have done, our findings indicate that the discontinuity caused by the transition of HIRS 2 to HIRS 3 is not completely removed when looking at the low $T_{12}$ values, so that further corrections

are needed in order to bring HIRS 3 levels down to HIRS 2 levels. By applying additional corrections to the low $T_{12}$ values, we come up with a more consistent intersatellite–calibrated $T_{12}$ time series with reduced errors at the low $T_{12}$ values due to the transition from HIRS 2 to HIRS 3, which can be further used for analyses of extreme UTHi cases.

In the following we first show how high values of UTHi and ice–supersaturation behave when the transition between the two HIRS instruments occurs. Then we discuss several refinements to the intercalibration (that is, we work on the data that

are already intercalibrated by Shi and Bates, 2011). A new procedure is devised and will be explained (section 2). A couple of simple results from the new method are presented, and the new method is discussed in comparison to more traditional methods (section 3). Finally our results are summarised, conclusions are drawn and an outlook on future research necessities and possibilities is given.

## 2 The intercalibration problem

### 2.1 Retrieval of upper-tropospheric humidity and ice supersaturation

When we used these intercalibrated data to set up a time series of the number of occurrences of cases with ice supersaturation we found a strong increase, seemingly coincident with the transition from HIRS 2 to HIRS 3 and this unwanted suprise led





us to check the intercalibration especially for the transition again. The check disclosed problems especially at the low end of channel 12 brightness temperatures, i.e. at those data that are characteristic for the supersaturation cases.

We believe that the intercalibration of Shi and Bates (2011) works well for the bulk of the data but not so well in the tails of the $T_{12}$ distribution. Recall that the intercalibration was based on monthly and zonal averages of $T_{12}$, in other words, on a distribution with clipped tails (as averaging eliminates extremes). It is appropriate to consider intercalibration as an exercise

in linear regression. With clipped tails, the regression sees only the central part of a distribution, however the tails could in principle change the regression coefficients quite substantially because of a leverage effect (the distance of tail values to the pivot at the mean value is evidently particularly large, that is, they have a large lever, see von Storch and Zwiers, 2001, sect. 8.3.18).

In order to make progress and avoid excessive averaging we consider daily averages of $T_{12}$ in $2.5° \times 2.5°$ grid boxes of the

30 to 70°N zone, similar to the data we have produced for the study in Gierens et al. (2014). We use all days with common operation between N14 (HIRS 2) and N15 (HIRS 3). In total we have 1004 common days (between 1 January 1999 and 7 April 2005) with 730473 measurements in the same box on the same day, that is, data pairs.

Figure 1 shows a scatter plot of randomly selected 2% of the data pairs for the upper–tropospheric humidity with respect to ice, UTHi. (Calculations have been done with all data). The abscissa shows values measured by N15, while the ordinate shows

differences of the N15–measured minus the corresponding N14–measured values. While these differences scatter around zero, there is a trend to increasing differences from low to high UTHi(N15). The least squares linear fit for the differences, $\delta$, as a function of the independent variable, $x$, has the following equation:

$$(\delta/\%) = -9.39 + 0.3015 \, (x/\%), \tag{1}$$

whose slope differs quite substantially from the ideal value of zero. The considered N14 data contain 636 records with UTHi>

100%, but 2739 records of N15 have UTHi> 100%. There are only 256 cases where both N14 and N15 show supersaturation in the same grid box and on the same day. In spite of the apparent tendency of N15 to show more supersaturation, the maximum values are equal, 113% for both instruments. These results suggest that the intercalibration of channel 12 must be improved if one is interested in high humidity cases and, in particular, in ice supersaturation.

## 2.2    Regression-based intercalibration

Let us make a step back and consider the brightness temperatures $T_{12}$ measured with the HIRS instruments. Figure 2 shows the two–dimensional histogram of $T_{12}$ pairs (i.e. $\{T_{12}(N15), T_{12}(N14)\}$) in 1 K resolution. Recall that these data are both intercalibrated by Shi and Bates (2011) to N12, and indeed the data pairs cluster nicely and symmetrically around the $y = x$ line (black dashed line, hereinafter simply referred to as "diagonal"), which demonstrates that the intercalibration was quite successful. This statement can be corroborated quantitatively, as both data sets have similar measures: mean values (standard

deviations) of about $(240 \pm 5)$ K, a total range from 228 to 265 K, similar quartiles and medians. In spite of this, the maximum of the joint distribution (violet pixels) is not centered on the diagonal axis. In particular at low $T_{12}(N15)$ there appears a tendency of N15 to display lower values than N14, leading to the observed surplus of supersaturation cases relative to N14. The diagonal



does not, therefore, represent the best (least squares) fit, that is, the regression–type intercalibration can be improved. Linear regression (black solid line) yields the following fit:

$$(y/\mathrm{K}) = 41.63 + 0.8292\,(x/\mathrm{K}), \tag{2}$$

with a slope that is not very close to unity and an intercept that differs quite substantially from zero. (Note that a quadratic fit is not required; it does hardly give an improvement). The stars in figure 2 represent the marginal means of $T_{12}(N14)$ per 1 K interval of $T_{12}(N15)$. These represent bin–wise mean differences which could be used for intercalibration as well. The difference between the regression and bin–wise correction is yet small.

For the moment this demonstrates that the intercalibration of the channel 12 brightness temperatures can be improved using common daily data for single grid cells instead of zonal/monthly averages. Whether this improvement is as well useful for the retrieval of upper–tropospheric humidity values has still to be shown.

If we perform the retrieval of UTHi (Jackson and Bates, 2001) for N15 using the regression–corrected values of the channel 12 brightness temperature, $\widehat{T_{12}}$, where

$$\left(\frac{\widehat{T_{12}}}{\mathrm{K}}\right) = 41.63 + 0.8292\left(\frac{T_{12}}{\mathrm{K}}\right). \tag{3}$$

then the resulting scatter plot of the corresponding values of UTHi is shown in figure 3 in the same format as in figure 1. It is obvious that the strong upward tendency of the differences with the N15-value is no longer present; the data pairs cluster almost symmetrically around the zero line (black line). The corresponding linear fit is:

$$(\delta/\%) = -3.354 + 0.09764\,(x/\%), \tag{4}$$

that is, the slope is very close to zero, as desired.

Yet unfortunately we must note that the range of UTHi (N15) is dramatically decreased at the high end and that all cases of supersaturation are eliminated when this kind of intercalibration is indeed applied. So ironically, instead of reducing the number of supersaturation cases in N15 data to a level given by the corresponding number of such events in N14 data, the new regression–based intercalibration eliminates all supersaturation. The comparison of this feature between N14 and N15 has in no way been improved, it has merely been turned upside–down. We note that similar procedures like bin–wise intercalibration with and without outlying data pairs (more than $\pm 3\sigma$ distance from the regression line) does only lead to minor modifications. The basic problem, that is, the strong elimination of high UTHi values and the complete loss of supersaturation, remains. Thus, the regression method, however a natural choice it might appear for the purpose of intercalibration, does not lead to plausible results. We need another procedure.

## 2.3 Intercalibration via the distribution function

The goal of the new intercalibration exercise is to have similar number of supersaturation cases for the data overlap period of N14 and N15, because the strong jump detected in the original data seems implausible even when one acknowledges that the





two satellites see the same grid cell at different times during a day. Looking at the cumulative distribution functions (cdf) of the corresponding channel 12 brightness temperatures (fig. 4) discloses the origin of the difference in supersaturation cases: there are much more (exceeding a factor of three) cases of very low $T_{12}$ values measured by HIRS 3 than by HIRS 2, a tendency that could already be observed in the 2–D histogram of fig. 2. As low $T_{12}$ produces high UTHi in the retrieval, this difference at the low $T_{12}$ tail produces the corresponding difference in the high UTHi tail.

We devised an alternative intercalibration procedure that yields similar distribution functions (with the N14 cdf as reference)
as follows: The data sets are grouped in $T_{12}$-bins first, the data in each bin are counted, resulting in numbers $n_t^s$ (where the upper index $s$ labels the satellite and the lower index $t$ the $T_{12}$ interval). We start with the lowest bin and compare $n_1^{15}$ with $n_1^{14}$. As there are more cases with low brightness temperature measured by N15, $n_1^{15} - n_1^{14} = \delta n_1 > 0$. Now we determine a minimimal temperature correction $\Delta T_1$ such that if all $T_{12}$ in the first bin of the N15 data set are incremented by this value, the surplus $\delta n_1$ of them get shifted to the next bin, and as a result, the first bin contains an equal number of data from N14
and N15, as desired. For the next bin we use the same procedure where we take into account the $\delta n_1$ additional values that have been shifted from the foregoing bin. The process is stopped when a bin is reached where either the ratio of the two cdfs is already close to unity or where this happens after the data from the bin below are shifted up. Note that we take the ratio between the cdfs, not their difference. This has the consequence that the corrections approach zero as the cdfs both approach unity, that is, the corrections are applied just at the low $T_{12}$ tail where we want to apply it; unnecessary corrections in the upper
bins are avoided.

What is the best bin width $\Delta$ for such a procedure? We could use Sturges' rule (or similar ones) to determine it:

$$\Delta \approx \frac{\max(T_{12}) - \min(T_{12})}{1 + \log_2 n} \tag{5}$$

which gives a $\Delta$ of approximately 1 K. Indeed the maximum correction $\Delta T_t$ is smaller than 0.8 K when a bin width of 1 K is chosen. If the bin width is smaller the necessary shifts get smaller as well, but at a low rate such that the maximum correction
can exceed $\Delta$, which means that some data would have to be shifted by more than one bin. This happens for $\Delta = 0.5$ K where the maximum shift computed exceeds 0.6 K. Shifting data by more than one bin would render the bookkeeping of shifted data unnecessarily complicated; thus we avoid it. The corrections for 1 K bins are shown in fig. 5 as blue dots. Comparison with the red regression line shows that the new corrections are much smaller than those determined by regression–like methods. The corrections are even zero above $T_{12} > 240$ K, due to the termination criteria of our algorithm.

The result of this kind of intercalibration for the intercomparison of the two brightness temperature data sets is shown in Fig. 6. Although the 2–D histogram is very similar to the one shown in Fig. 2, there are notable differences. The gravity centre of the joint distribution (violet pixels) is now following the diagonal axis (dashed black line), a desired feature. The best linear fit (solid black line) is still tilted against the diagonal; its equation is

$$(y/\mathrm{K}) = 29.89 + 0.8771\,(x/\mathrm{K}). \tag{6}$$

The intercept is much smaller than for the original data, and the slope is a bit closer to unity than before. Marginal means of $T_{12}(N14)$ (stars) again closely resemble the linear regression. The marginal means and the regression are very close to the $y = x$ diagonal in the gravity centre of the distribution.





The result of the cdf–based intercalibration is shown for UTHi in Fig. 7. It is seen that high and supersaturation values of UTHi are retained, as desired. The price for this is however that the regression between the N15 and the N15 vs N14 difference has a higher slope than with the regression–based intercalibration (Fig. 3). The linear equation is:

$$(\delta/\%) = -7.566 + 0.2068\,(x/\%). \tag{7}$$

Taking $x = 100\%$ (saturation), this gives a deviation of about 13% between the two data sets. This is a moderately large
uncertainty that must be accepted if one is interested in a long time series of high UTHi and supersaturation cases.

It is not necessary to show the $T_{12}$ cumulative distribution functions after the correction; these are almost equal *qua* construction.

## 3   Results and discussion

### 3.1   Overall improvement

Simple statistical measures, computed with the set of the common daily and grid based data, may show that indeed an improvement results from the cdf–based intercalibration. The indicators are the following:

- The mean difference of channel 12 brightness temperature (N15 minus N14) is $(-0.63 \pm 2.76)\,\mathrm{K}$ in the original data (mean and one standard deviation). With the correction applied to N15 it reduces to $(-0.35 \pm 2.70)\,\mathrm{K}$.

- The mean difference of the corresponding UTHi is $(3.24 \pm 12.41)\%$ in the original data. With the correction it reduces
to $(0.54 \pm 11.50)\%$.

Thus the mean temperature difference is almost halved, the mean UTHi difference is even reduced by a factor of six.

### 3.2   Simple applications

For testing the procedure further we consider the 256 data records indicating ice supersaturation in both measurements (N14 and N15). These pairs of brightness temperature and UTHi are shown in Fig. 8 with black points showing the original values
and red points the modified ones, after application of the cdf–based intercalibration. All N15-brightness temperatures of these cases are shifted to slightly higher values and thus all corresponding UTHi values are decreased. 176 of the cases (more than two thirds of them) change from supersaturated to subsaturated in the N15 data, but all remain at above 90%, that is, they still indicate quite moist conditions.

Figure 9 shows 35–year time series of UTHi threshold exceedances. This is the fraction of data with UTHi$\geq X\%$, where $X$
is 70, 80, 90, and 100. This counting exercise has been performed with the original data (shown in the upper panel) where a strong increase in high UTHi cases can be observed from about 1999 onwards for all selected thresholds. Although it looks like an indication of climate change there is none. A similar analysis with the modified data shows no obvious signs of a trend and it will need sophisticated time-series-analytical methods to find out if there are any trends in the data at all. A deeper analysis of the four time series will be reported in a forthcoming paper.





### 3.3 Discussion

Actually there are two questions to be discussed:

– Is it justified at all to combine all HIRS $T_{12}$ data into a single time series when it is a matter of fact that HIRS 2 and HIRS 3/4 sense different layers of the upper troposphere, layers that overlap heavily but whose centres are more than one kilometre apart vertically?

– Is it justified to use a cdf–based intercalibration procedure?

The first of these questions is a difficult one; and it is just the basic question of a number of subsequent problems as, for instance, under which circumstances is it justified or not? Which assumptions have to be made about the structure of temperature and moisture profiles, etc. This technical note is not the place to answer these questions; but it certainly deserves much more research in order to be sure that results obtained sofar (Gierens et al., 2014; Chung et al., 2016) are reliable. This should be a topic for the near future.

To discuss the second question needs an analysis and comparison of what is effectively done in the cdf–based and the regression–like methods. It should be noted that the only subjective element in the intercalibration problem is the choice of the method. Once the method has been chosen, everything else is based on fixed rules and is therefore objective. The difference in the methods is the different set of rules and the reasoning from which these rules are derived. In the end, the procedures are similar again: All methods are used to determine a $T_{12}$-dependent correction which is then applied.

– The regression method is based on the postulate that the mean squared difference between *all* data pairs is a minimum (regression of the second kind).

– The method of Shi and Bates (2011) is based on the postulate that the mean squared difference between data pairs in given intervals (bins) of $T_{12}$ is a minimum (regression of the first kind). This method is more flexible than the regression-based method since it does not assume a linear relation between the two data sets. As one can see in fig. 2 (black line and stars), both methods give very similar results.

– The cdf–based method is based on the postulate that $P\{\widehat{T_{12}}(N15) \leq T\}/P\{T_{12}(N14) \leq T\} \approx 1$ ($P\{\cdot\}$ is the probability of the event stated in the brackets), i.e. that both cumulative distributions are similar.

There might be further possibilities which can be based on still other postulates. For instance, instead of considering the relative differences between the two cdfs one could as well use the absolute differences and postulate that these are close to zero. To our knowledge there is no principle argument favouring one or another of these.

One essential difference between regression–based and cdf–based methods is that the first consider the data as pairs while this connection is given up in the latter method. The latter instead considers the statistical properties of the data as two independent populations. The reasoning for that has pros and cons. Considering the data as pairs is justified to a certain degree since they are taken on the same day in the same grid cell. But they are also taken at different times of the day which loosens the connection.



Since the truth is unknown nobody can decide which method gives results closer to reality. However, it would be very implausible that supersaturation would suddenly occur much more frequently than before (original data), or not anymore (regression-based methods). If the NOAA HIRS channel 12 time series can be combined at all (a question not to be solved here) we need an intercalibration that keeps a certain level of supersaturation frequency and the most conservative choice is then that a change of the UTH distribution functions during the 1004–day transition period from one to the next satellite should be small. So for us it was simply a practical decision guided by this conservative assumption to choose the cdf–based method.

Further evidence for choosing the cdf–based method, as a plausible intercalibration method to account for values found at the low tail of $T_{12}$ distribution when it comes to analyse high UTHi values, is provided in Table 1, which shows the average fraction of UTHi exceedances from 70% to 100% during three periods of interest: the period before the transition from HIRS 2 to HIRS 3/4 (1980–1999), the period during the transition (1999–2005) and the period after the transition to HIRS 3/4 (2006–2014). The table also shows the mean fraction of exceedances before (a) and after (b) the corrections applied based on the cdf–

method, together with the differences of the means between (b) and (a), i.e. cdf–corrected data minus original data, indicative of improvements performed in the original data. All averages and corresponding differences are expressed in percent.

For the case of 70% UTHi threshold, the original data suggest that the mean fraction of exceedances increased from about 1.6% in the period 1980–1999 to about 3.8% in 2006–2014, corresponding to an overall increase of about 138% within about two decades or so. The respective changes for the cases of 80%, 90% and 100% UTHi thresholds by the original data were even

larger. Although the mean fraction of exceedances is generally small for the examined UTHi thresholds, such large changes from one period to another do not sound reasonable and are indicative that something maybe wrong in the data. Application of the cdf–based correction to the UTHi threshold data of 70% reduced the change from 138% to 9%. Significant improvements were also found at the other UTHi thresholds. The differences between the cdf–corrected data and the original data in the periods examined are obvious (Table 1c). Our findings suggest that extreme UTHi cases might have increased in the past 35

20 years. However, given that the zonal mean UTHi remained almost unchanged during the period 1979–2014 (Chung et al., 2016) it is doubtful whether the observed changes estimated with the original data are real. The observed changes estimated with the cdf–based method (Table 1b) look more reasonable than those calculated with the original data (Table 1a).

## 4 Conclusions

We developed a new method for intercalibration of satellite data that is based on a comparison of distribution functions of

25 brightness temperatures instead of regression methods. We applied this intercalibration to channel 12 brightness temperatures measured with the HIRS 2 instrument on NOAA 14 and the HIRS 3 instrument on NOAA 15. These data have had already been intercalibrated by Shi and Bates (2011) but there were still discrepancies at the low end of the distribution, perhaps a consequence of basing their intercalibration on monthly and zonal means which can smooth extremes away. Here we based our additional intercalibration on daily data in $2.5° \times 2.5°$ grid boxes. The originally intercalibrated data show a very strong increase

of very low brightness temperatures with the transition from HIRS 2 to HIRS 3, and this translates into a correspondingly strong




increase in the frequency of occurrence of ice supersaturation in upper–tropospheric humidity with respect to ice retrieved from the brightness temperatures. This seemed to us unphysical and implausible.

We tried regression–based intercalibration procedures first but without success. Instead of less ice supersaturation in HIRS 3 data, all supersaturation cases were eliminated because the corrections were too large. This again seemed to us unphysical and implausible.

The new intercalibration method is constructed in a way that the probability of supersaturation does not change in the transition from HIRS 2 to HIRS 3. Of course, we do not know whether this assumption is correct; it is simply the most conservative assumption. Other data sets for the transition period (1999–2005) are needed to check the validity of this assumption. This is beyond the scope of the present paper.

The overall discrepancies between the $T_{12}$ data pairs of HIRS 2 on NOAA 14 and HIRS 3 on NOAA 15 is reduced when the new intercalibration is applied. The mean difference in terms of brightness temperature is almost halved, and the mean difference of the retrieved UTHi is even reduced by a factor of six.

A fundamental question is whether and under which conditions HIRS 2 and HIRS 3 data can be combined into a single time–series at all, since they sense different layers in the upper troposphere. For the present investigation we have assumed, as a working hypothesis, that such a combination is admissible. It is not in the scope of this paper to begin a investigation of this difficult problem, but it is certainly a topic for the next future.

## 5 Code availability

IDL code for the cdf nudging can be obtained from the first author on request.

## 6 Data availability

HIRS data in general are available from NOAA. The data used for the present paper can be obtained from the authors on request.

*Author contributions.* KE provided the raw UTHi and BT data, KG wrote and ran the IDL codes. Both analysed the results and wrote the text.

*Competing interests.* The authors declare that they have no conflict of interest.

*Acknowledgements.* We thank Dr. Shi (NOAA) for providing 35 years worth of HIRS brightness temperature data and for her help and advice when problems occurred. Christoph Kiemle and Robert Sausen read a manuscript version of the paper and helped to improve it.





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



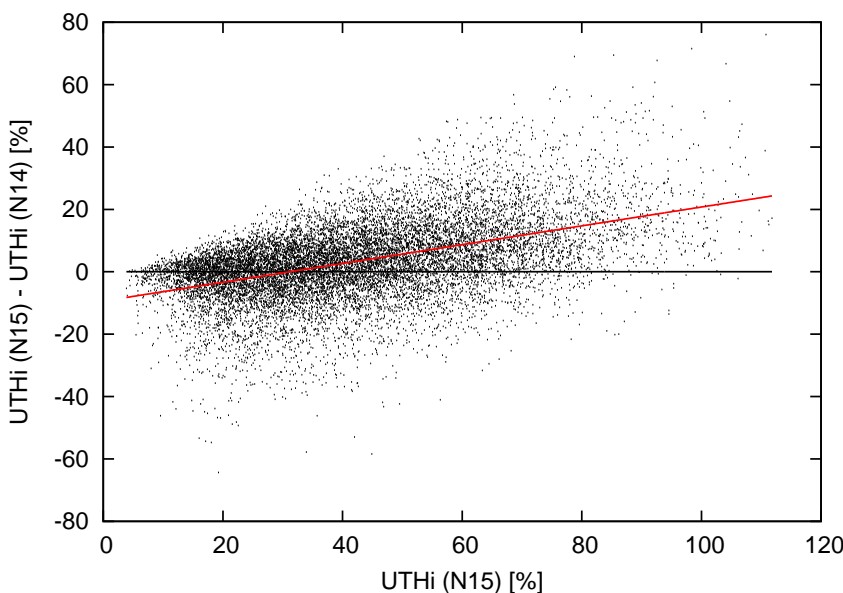

**Figure 1.** Scatter plot of data of upper-tropospheric humidity with respect to ice (UTHi, in percent), retrieved from channel 12 brightness temperatures from the HIRS 3 instrument on NOAA 15 and from the corresponding HIRS 2 instrument on NOAA 14. The data pairs represent daily average values taken in $2.5° \times 2.5°$ grid boxes in the northern latitude belt of 30 to 70 °N. The ordinate displays the respective differences which should ideally scatter symmetrically around the zero line (black solid line). A remarkable upward trend of the data cluster with increasing UTHi (N15) is evident; a corresponding least squares fit (red line) has an upward slope of about 0.3.





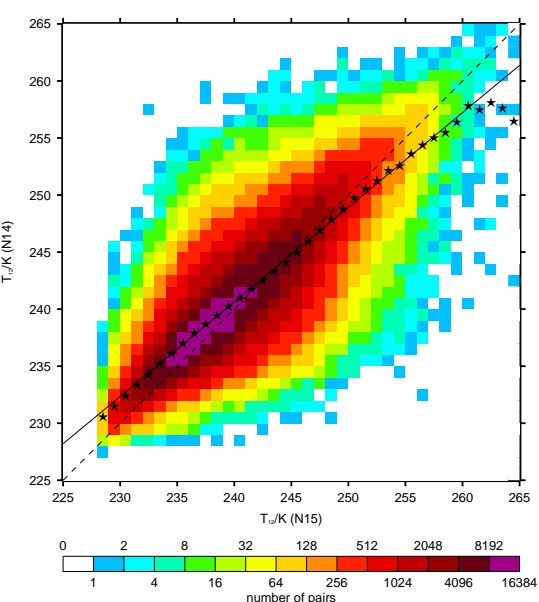

**Figure 2.** Two-dimeansional histogram of $\{T_{12}(N15), T_{12}(N14)\}$)-pairs in 1 K resolution. Ideally the gravity centre of the joint distribution (violet pixels) would follow the diagonal axis (dashed black line), but it is slightly shifted above the axis. The best linear fit is given by the solid black line. Marginal means of $T_{12}(N14)$ for each 1 K interval of $T_{12}(N15)$ are represented by stars; they closely resemble the linear regression. Both show that $T_{12}$ measured by HIRS 3 is underestimated at the low end of the data range which causes an excess of supersaturation in the UTHi retrieval.





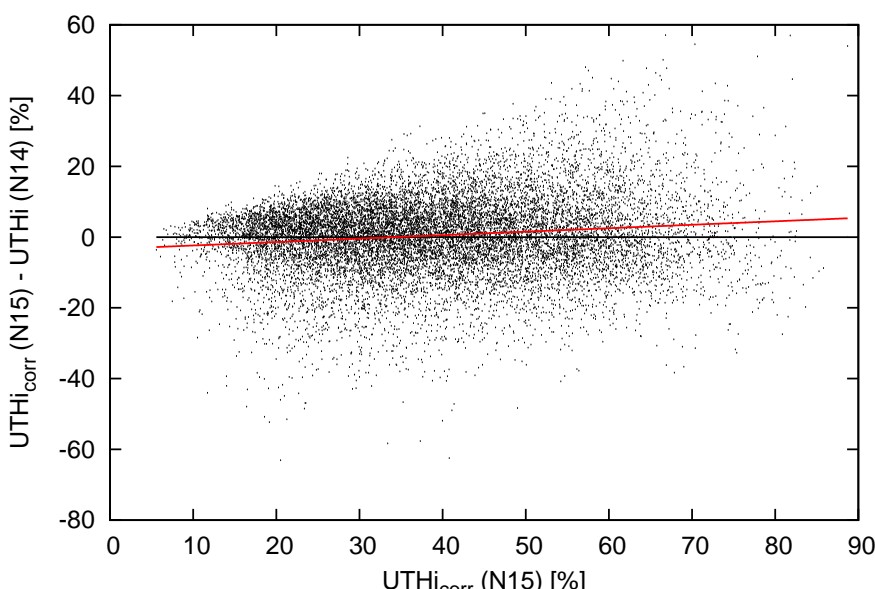

**Figure 3.** As Fig. 1, but with UTHi retrieved from modified channel 12 brightness temperatures for HIRS 3 on NOAA 15, according to eq. 3. The data nicely scatter around the $x = 0$ line (black). Yet the best fit is still slightly tilted against the horizontal (red line), a minor problem. The big problem is that the range of UTHi is drastically reduced to values slightly exceeding 90%, that is, instead of reducing the number of supersaturation cases, they have been eliminated, an undesired effect.



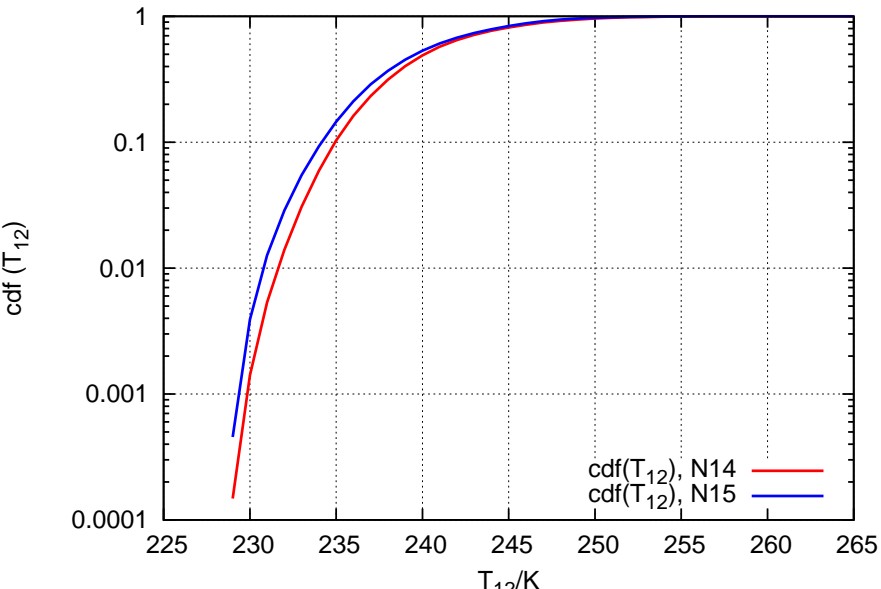

**Figure 4.** Cumulative distribution functions (cfd) of channel 12 brightness temperatures, measured with HIRS 2 on N14 (red) and with HIRS 3 on N15 (blue). Note the quite large discrepancy (in relative terms) between both cdfs at low values of $T_{12}$.




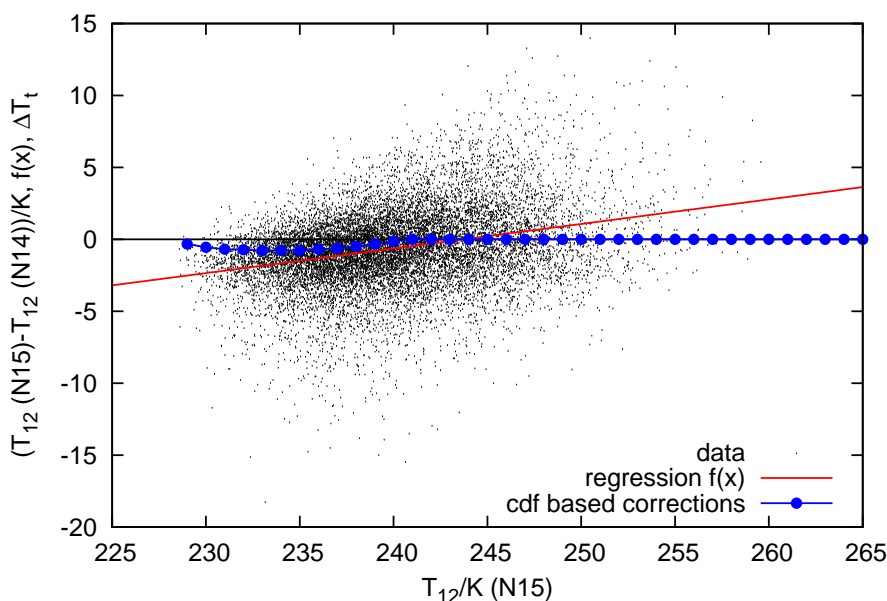

**Figure 5.** As Fig. 2, but with the brightness temperature difference on the ordinate axis. The red regression line corresponds to the red line in Fig. 1. The blue points are the intercalibration corrections determined for 1 K bins using the procedure described in the text. Note that this procedure leaves all data exceeding 240 K unchanged and that the necessary corrections at lower brightness temperatures are smaller than the regression based corrections.





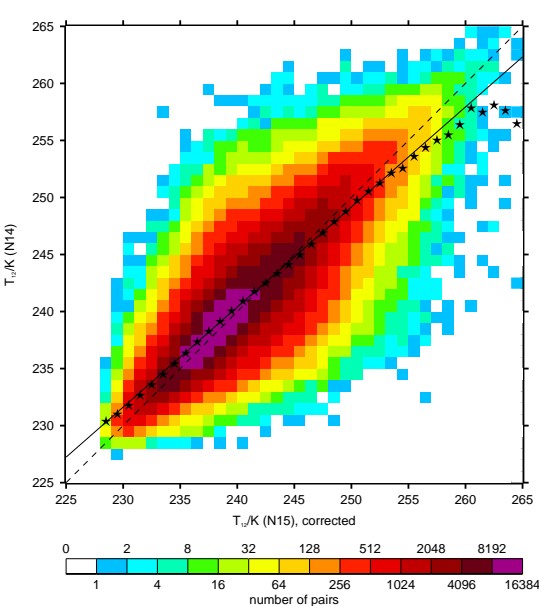

**Figure 6.** As Fig. 2, but after correction of $\{T_{12}(N15)$ with the cdf-based procedure described in the text. The gravity centre of the joint distribution (violet pixels) is now following the diagonal axis (dashed black line), however the best linear fit is still tilted against the diagonal; it is given by the solid black line. Marginal means of $T_{12}(N14)$ for each 1 K interval of the corrected $T_{12}(N15)$ are represented by stars; they again closely resemble the linear regression. The tilt between the best fit and the diagonal is smaller than in Fig. 2, which means that the correction reduces $T_{12}$ (HIRS 3) overestimation relative to $T_{12}$ (HIRS 2).



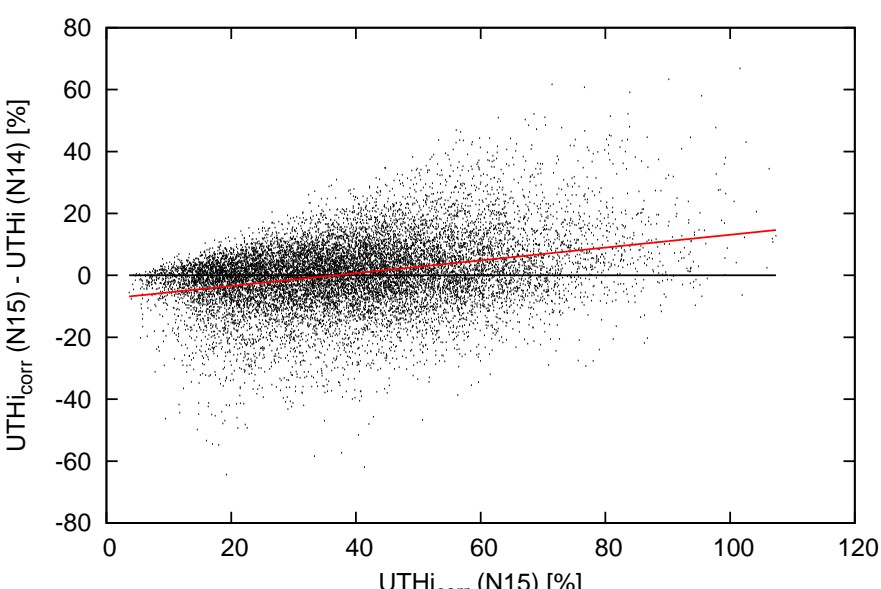

**Figure 7.** As Fig. 3, but with intercalibration via the cumulative distribution function of brightness temperatures. While this procedure leaves supersaturated cases in the N15 data set, the linear correspondence between N14 and N15 UTHi values is worse than with a regression based intercalibration (compare the slopes of the red lines here and in Fig. 3).





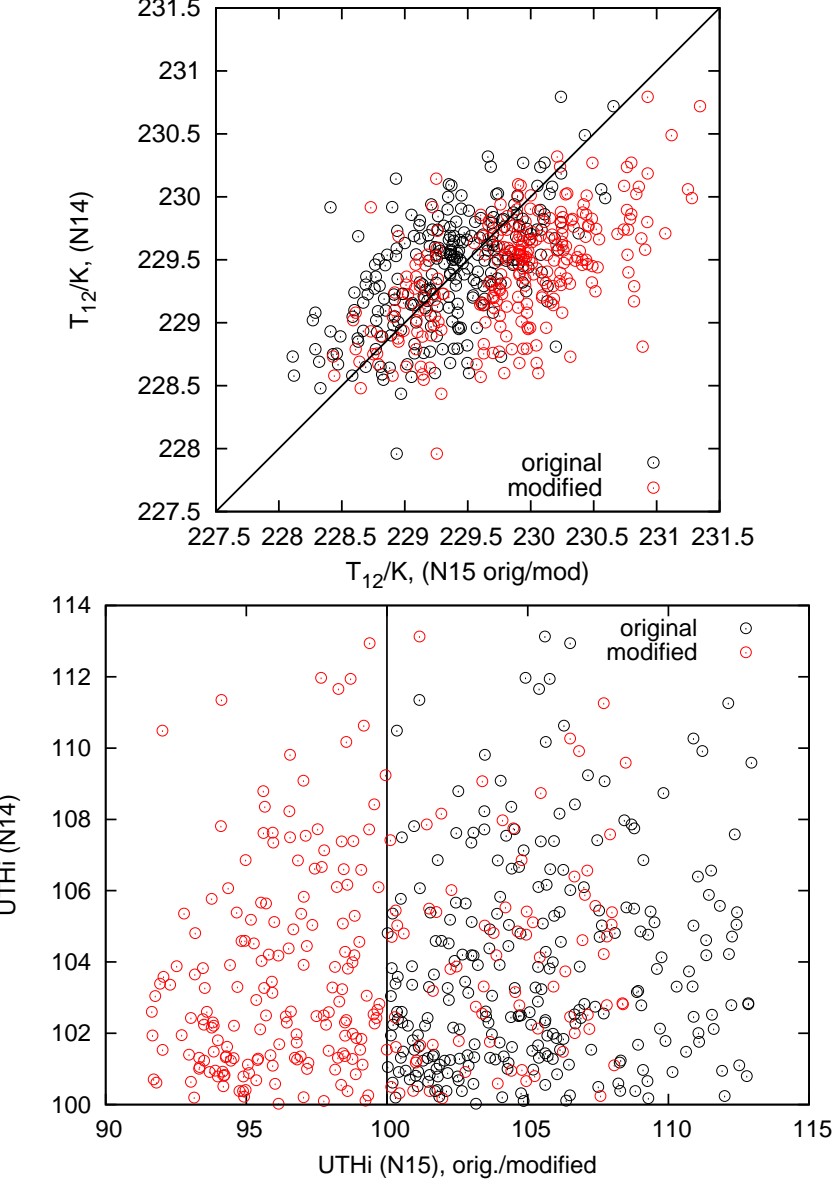

**Figure 8.** 256 data pairs where both satellites report ice supersaturation (black points) and their modification after application of the cdf–based intercalibration (red points). Top: Effect of the modification of the N15–measured brightness temperature. Bottom: Effect of the modification on UTHi. More than two thirds of all N15–supersaturation cases are shifted to a UTHi value between 90 and 100%.



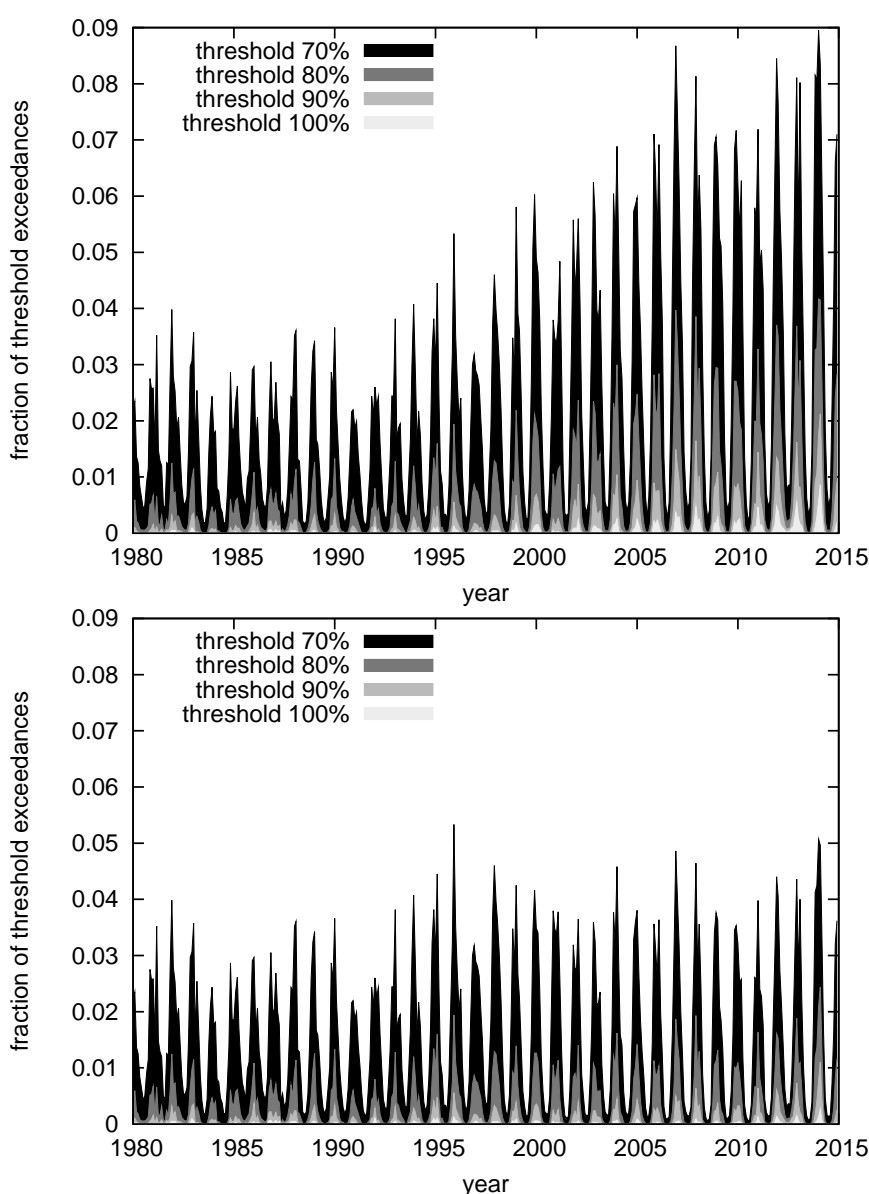

**Figure 9.** Raw time series of fraction of exceedances for UTHi thresholds from 70% to 100% before (top) and after (bottom) application of the cdf–based $T_{12}$ intercalibration for all satellites beginning from N15. The data until 1998 are identical in both panels, but note the different ordinate scales. The raw time series after correction (bottom) does not show peculiar jumps and sudden increases anymore.





**Table 1.** Average fraction of exceedances and corresponding standard deviations (both in percent) for UTHi thresholds from 70% to 100% during 1980–1998 (period before the transition from HIRS 2 to HIRS 3/4), 1999–2005 (transition period) and 2006–2014 (post–transition period), before (a) and after (b) application of the cdf–based $T_{12}$ intercalibration. (c) shows the differences of the means between (b) and (a).

|  | 1980–1998<br>(pre–transition period) | 1999–2005<br>(transition period) | 2006–2014<br>(post–transition period) |
|---|---|---|---|
| | (a) before cdf–based corrections | | |
| 70% | $1.59 \pm 1.11$ | $2.85 \pm 2.03$ | $3.80 \pm 2.70$ |
| 80% | $0.33 \pm 0.36$ | $0.83 \pm 0.84$ | $1.31 \pm 1.25$ |
| 90% | $0.07 \pm 0.10$ | $0.21 \pm 0.28$ | $0.40 \pm 0.49$ |
| 100% | $0.01 \pm 0.02$ | $0.04 \pm 0.07$ | $0.10 \pm 0.15$ |
| | (b) after cdf–based corrections | | |
| 70% | $1.59 \pm 1.11$ | $1.68 \pm 1.37$ | $1.73 \pm 1.50$ |
| 80% | $0.33 \pm 0.36$ | $0.41 \pm 0.47$ | $0.51 \pm 0.58$ |
| 90% | $0.07 \pm 0.10$ | $0.09 \pm 0.12$ | $0.14 \pm 0.20$ |
| 100% | $0.01 \pm 0.02$ | $0.01 \pm 0.02$ | $0.02 \pm 0.04$ |
| | (c) differences between (b) and (a) | | |
| 70% | 0 | $-1.17$ | $-2.07$ |
| 80% | 0 | $-0.42$ | $-0.80$ |
| 90% | 0 | $-0.12$ | $-0.26$ |
| 100% | 0 | $-0.03$ | $-0.08$ |