# Peer review of "Technical Note: On the intercalibration of HIRS channel 12 brightness temperatures following the transition from HIRS 2 to HIRS 3/4 for ice saturation studies"

_Atmospheric Measurement Techniques, 2016_

## Referee Comment (RC1) · Anonymous Referee #3 · 3 Dec 2016

Major comments

This paper demonstrates faults with a regression-based approach for intercalibrating HIRS Ch. 12 between HIRS version 2 and HIRS version 3. While it is arguably questionable to intercalibrate the 6.5 micron and and 6.7 micron channels, which sample such different layers of the atmosphere, the authors are straightforward about this, and they are not the first to attempt to do so.

The authors are interested in near- and super-saturated relative humidity with respect to ice. This is at the tail end of the distribution of brightness temperatures intercalibrated

by Shi and Bates, so it is unsurprising that their intercalibration method underperforms in this case.

The authors go on to demonstrate the failures of a linear regression approach, and instead suggest a cumulative frequency distribution (CFD)-based approach. This approach seems to make the corrections they seek, while leaving the rest of the values more-or-less alone.

I have no problems with the CDF approach the authors chose. But I would like to see it compared to something that doesn't fail as terribly as the OLS regression line. Glancing at figure 2, it can be seen that the regression curve is flat, thanks to regression dilution. (See Pitkänen et al., 2016; doi:10.1002/2016GL070852). The elimination of super-saturation upon its application is a direct result. Rather than continuing with their critique of the linear regression method - with fails almost trivially - some other standard technique ought to have been applied. May I suggest calculating instead a bivariate regression (see York reference in in Pitkänen)? Practically, this can be done by choosing a line that goes through the center of mass of the scatter plot, with the same slope as the first eigenvector of the 2x2 covariance matrix of the 2xN time series of pairs. The first eigenvector points in the direction of maximum variance, thus minimizing the residuals perpendicular to the line, rather than in an arbitrarily chosen y-direction.

Also, I think the authors should be more clear how they choose pairs of data points. For example 2 HIRS/2 points "A" and "B", and 2 HIRS/3 points "1" and "2", if all close together, can produce 4 pairs for comparison: A-vs-1, A-vs-2, B-vs-1, and B-vs-2. Do the authors avoid this sort of inflation?

Several figures are plotted with very fine points which only appear at certain zooms in Adobe Acrobat; I suggest the authors use larger points that are semi-transparent, or use a heat map. Also, the rainbow color scheme chosen is neither perceptually uniform nor is it color blind safe.

The language and writing are understandable, but not publication-ready. The paper could use some revision for language.

Some minor comments follow:

p1 7, "We present that" should be "We show that" 21, "Ticklish" is informal

p2 1-3, "Relatively few papers..." Can you cite some? It looks like you do later; Gierens 2014, Lamquin et al., 2009, Dickson et al., 2010 5, "to study" should be "with which to study"

p3 27-28 In which section?

p4 29-30, Mean value is in parenthesis with the standard deviation, contrary to the use of parenthesis beforehand.

p7 26-27, "Although it looks like an indication of climate change there is none." This statement is sweeping. (Of course there is climate change all the time. That's just probably not what we're looking at.)

Figure 9, the scales are all the same, contrary to the caption

---

## Referee Comment (RC2) · Anonymous Referee #2 · 15 Dec 2016

In this study, the authors examined time series of the intercalibrated HIRS brightness temperature data at channel 12 (T12), and found a discontinuity in the time series of ice supersaturation inferred from T12 during the transition period from the HIRS 2 to the HIRS 3 instrument, which involved a shift of the central wavelength in the channel. To correct for this apparent discontinuity at the low end of T12, the authors proposed an additional correction method based on the cumulative distribution functions of T12 from NOAA-14 and -15 satellites. The proposed correction method is expected to help remove large part of the bias at the low end of T12. The authors may consider the following issues to further demonstrate the robustness of the conclusions.

Cloud contamination of infrared measurements can cause a positive bias in the upper tropospheric humidity estimated from T12 observations. This means that differences in the cloud clearance between HIRS 2 and HIRS 3/4 may contribute to the discontinuity in the time series. As a result, it is important to make sure that the cloud clearance at the low end of T12 is consistent between HIRS 2 and HIRS 3.

Due to the diurnal variations of cloud ice mass and humidity in the upper troposphere, difference in the local observation time between satellites may lead to discrepancy in the observed brightness temperatures. In addition, the orbit of NOAA-14 has substantially drifted during the transition period. Have potential biases arising from these factors been taken into account here?

The proposed intercalibration method is based on the assumption that the probability of supersaturation did not change during the transition period from the HIRS 2 to the HIRS 3 instrument. The validity of this assumption can be assessed using microwave observations (e.g., Buehler et al., 2008, JGR) or the free-tropospheric humidity data set constructed from the Meteosat MVIRI and SEVIRI observations (e.g., Schroder et al., 2014, ACP).

L400-410, Fig. 8: This portion is confusing because the data pairs of the unmodified values show better agreement than the pairs of the modified values.

L515-519, Table 1: The mean fraction of exceedances is very small for the examined UTHi thresholds. This gives an impression that it might be okay not to correct for the discontinuity at the low end of T12.

Fig. 1: To further demonstrate that the discontinuity during the transition period is caused by the shift of the central wavelength, the authors may also present scatter plots for matching pairs between NOAA-12 and NOAA-14 and between NOAA-15 and NOAA-16.

Fig. 3: the y = 0 line instead of the x = 0 line?

Fig. 4: "cfd" needs to be replaced by "cdf"

Fig. 5: As Fig. 1 instead of As Fig. 2?

Fig. 9: The scales are all the same.

—————————————————

---

## Author Comment (AC1) · 31 Jan 2017

**Reply to the reviewer's comments on Manuscript No. AMT-2016-289 by K. Gierens and K. Eleftheratos**

**Remarks**

We thank all reviewers for their very constructive comments which helped improving the paper. Below you find a detailed response to all questions raised. Reviewer comments are reprinted in italics font, our replies in times roman.

According to the replies presented below we have revised our manuscript. You will find passages with substantial changes marked in red in the revised version.

Here we would like to start with a comment on the significance of this research which was rated fair by two reviewers. We believe that this assessment is wrong. In the revised paper we have added arguments that show why such studies as this one are important.

As we state in the paper, the original intercalibration by Shi and Bates (2011) is quite successful for the bulk of the channel 12 data. That is, if the mean of  $T_{12}$  or of UTHi is relevant, the intercalibration is sufficient. However, the mean means nothing in non-linear processes like radiation and cloud formation. This implies that more than just the first (and perhaps the second) moment of the UTHi distribution is needed, in particular characteristics of the tails of its distribution. It is clear that for cloud research and how cloudiness will change with climate change, information on the upper tail of UTHi is needed. For questions of the radiative balance of the Earth it is important how the very dry regions of the subsidence zones (termed "radiator fins" by Pierrehumbert, 1995; see also Schröder et al. 2014 and Roca et al. 2011) behave with ongoing climate change; thus the dry end of the UTHi distribution is of immense interest as well. These arguments show that homogeneous time-series of the whole UTHi distribution are needed, it is not sufficient that just the time-series of the mean is smooth. From these considerations we believe that technical studies like ours are not only of marginal interest. They are important to improve the quality of the time-series for the whole of the UTHi distribution.

These arguments are now included as the last paragraph in the discussion section 3.3 of the revised manuscript.

**1 Review-independent major changes**

In the course of the revision process we noticed that it makes no sense to compute a regression line for a plot like in our old figure 1. The reason is simple, but the problem has not been noted by anyone so far. As UTHi cannot be negative, the difference UTHi(N15)–UTHi(N14) strongly tends to negative values (i.e. UTHi(N14)>UTHi(N15)) when UTHi(N15) is small. Given that UTHi(N15) is small, it is quite improbable that UTHi(N14) is even smaller. At the other end of the distribution we have a similar phenomenon, as values exceeding 115% do not occur in our data sets. Thus, given that UTHi(N15) reaches the upper extreme, it is much more probable that UTHi(N14) remains smaller than that it would be even larger. This means that, unless all data pairs agree perfectly, a scatter plot like that in figure 1 *must* have a rhombic shape with an surplus of negative ordinate values. It is clear that a linear fit through a such shaped cloud of data points must have a positive slope. It might be that the slope depends on the width (standard deviation) of the individual distributions but our statement that it "differs quite substantially from the ideal value of zero" is, albeit true, meaningless for the problem at hand.

Therefore we replace such plots by simple y vs x plots, see the new figure 1 in the revised version. In such a plot the problem becomes evident through an unequal number of points above

and below the y = x diagonal line.

Similar changes apply to figures 3 and 7.

The old scatterplots were not wrong but their regression lines were misleading. They are appropriate to perform a bin-wise regression (regression of the 1st kind) but not for a regression of the 2nd kind. We have redrawn all figures in typical x-y form, which is typically used to study the linear correlation between two parameters.

Figure 5 has been completely revised. It has been reduced to show the corrections and their actual values more clearly, but not the data points already shown before which is not necessary.

**2 Reply to reviewer No. 2**

**2.1 Cloud contamination**

Cloud contamination of infrared measurements can cause a positive bias in the upper tropospheric humidity estimated from T12 observations. This means that differences in the cloud clearance between HIRS 2 and HIRS 3/4 may contribute to the discontinuity in the time series. As a result, it is important to make sure that the cloud clearance at the low end of T12 is consistent between HIRS 2 and HIRS 3.

Note that we have used the intercalibrated data of Shi and Bates (2011). These data are cloud-cleared as stated in their paper (page 3, beginning of par. 9):

"The HIRS data are first processed to remove cloudy pixels for the water vapor field. The cloud clearing procedure follows the method detailed by Jackson et al. [2003]. The process is accomplished using a simplified method based on the ISCCP cloud detection approach [Rossow and Garder, 1993]."

We add a statement on this in section 3.2.

**2.2 Diurnal variations**

Due to the diurnal variations of cloud ice mass and humidity in the upper troposphere, difference in the local observation time between satellites may lead to discrepancy in the observed brightness temperatures. In addition, the orbit of NOAA-14 has substantially drifted during the transition period. Have potential biases arising from these factors been taken into account here?

Indeed, diurnal variations of relative humidity in the upper troposphere could lead to spurious variations in the NOAA channel 12 time series. But such effects would occur not just related to the HIRS/2 to HIRS/3 transition. The NOAA series is a succession of morning and afternoon satellites. Thus, if systematic diurnal humidity variations have indeed an effect on the time series, this effect must be present since N7 started. We have looked at the cumulative probability distributions of UTHi (as well as T12 and T6) for all satellites and all years. We find essentially two groups of curves: (i) N6 to N14, (ii) N15 and all later satellites, see Fig. 1 at the end of this reply. Please note the gap between the dark brown (N14) and the violet (N17) curves. The cdf curves of all satellites before N14 are left of the N14 curves while all satellites of the HIRS3/4 era are to the right. There is nothing in these curves that suggests effects of diurnal variations, since then we would expect two other groups, namely one containing morning satellites and the other containing afternoon satellites. A strong orbital shift would then smear out the distinction between these hypothetical groups.

It is however remarkable that the N14 cdf curves are at the right edge of the HIRS2 group. The reason for this and whether it results from a fact other than typical interannual variability is unknown and a topic for future research.

**2.3 Assumption that supersaturation did not change**

The proposed intercalibration method is based on the assumption that the probability of supersaturation did not change during the transition period from the HIRS 2 to the HIRS 3 instrument. The validity of this assumption can be assessed using microwave observations (e.g., Buehler et al., 2008, JGR) or the free-tropospheric humidity data set constructed from the Meteosat MVIRI and SEVIRI observations (e.g., Schroder et al., 2014, ACP).

This is indeed a working hypothesis that was necessary to do the correction. Of course the frequency of supersaturation might have changed over time, which is not known and which is a reason for our studies. It is however very implausible that it has changed such dramatically just at the transition to HIRS/3. The increase of the frequency of threshold exceedances is not small, it is more than a 3-sigma increase when we compute the sigma from the first ten years of the time series. It is hardly conceivable that such a dramatic change could have happened unnoticed in other variables (for instance frequency and coverage of persistent contrails). Such changes have, at least to the author's knowledge, never been reported. Gierens, Eleftheratos and Shi (2014) indeed found a small decadal increase of UTHi in large regions of the northern mid-latitudes using the intercalibrated HIRS data. These decadal changes refer to the whole range of UTHi, not just the high humidity cases. It might be that the probability density function of UTHi has changed such that high humidities had experienced a stronger increase than the bulk of the distribution. These questions are not yet solved and their solution needs much more research (including analyses of the microwave data mentioned by the reviewer). This research is far beyond the topic of the current paper.

**2.4 Minor comments**

**2.4.1 L400-410, Fig. 8**

This portion is confusing because the data pairs of the unmodified values show better agreement than the pairs of the modified values.

We agree with the referee's observation. It is indeed the case that for these 256 data pairs the statistics is worse after the correction than before. Insofar, this selection led to a somewhat unfortunate example. However, there are much more cases, as stated, where N15 shows supersaturation while N14 does not. For these cases, the statistics is better after the correction. In the revised version we will add a comment similar to what we write here, but we will retain the somewhat "bad" example, since it shows honestly that every "bulk" correction (i.e., a correction that is not point by point) inevitably has its pros and cons.

**2.4.2 L515-519, Table 1**

The mean fraction of exceedances is very small for the examined UTHi thresholds. This gives an impression that it might be okay not to correct for the discontinuity at the low end of T12.

Indeed, the mean fraction of exceedances is small for the examined UTHi thresholds. However, we cannot ignore the fact these small values changed artificially following the transition from HIRS 2 to 3. As we focus on ice saturation and supersaturation cases and since we know what caused this unnatural discontinuity in the time series, it is important for us to find and apply methods that take care of this problem. Our method (cdf-based intercalibration) indicates that it is necessary to correct for the discontinuity at the low end of T12, when it comes to assess extreme UTHi values as in our case, and appears to solve the problem satisfactorily.

**2.4.3 Fig. 1**

To further demonstrate that the discontinuity during the transition period is caused by the shift of the central wavelength, the authors may also present scatter plots for matching pairs between NOAA-12 and NOAA-14 and between NOAA-15 and NOAA-16.

We refer back here to Fig. 1. The grouping of the cdfs is evidently according to the version of the HIRS instrument used on the various satellites. It is also clear from the plot that there are interannual variations and perhaps remaining sensor differences. Scatter plots of N12 vs. N14 and N15 vs N16 would lay the focus on these other differences which would be inappropriate as these are not the focus of the current paper.

**2.4.4 Figs. 3, 5**

The figures have been replaced and the problems don't not exist anymore.

**2.4.5 Figs. 4, 9**

All corrections done.

**3 Reply to reviewer No. 3**

**3.1 Consideration of regression dilution**

I have no problems with the CDF approach the authors chose. But I would like to see it compared to something that doesn t fail as terribly as the OLS regression line. Glancing at figure 2, it can be seen that the regression curve is flat, thanks to regression dilution. (See Pitkänen et al., 2016; doi:10.1002/2016GL070852). The elimination of super-saturation upon its application is a direct result. Rather than continuing with their critique of the linear regression method - with fails almost trivially - some other standard technique ought to have been applied. May I suggest calculating instead a bivariate regression (see York reference in in Pitkänen)? Practically, this can be done by choosing a line that goes through the center of mass of the scatter plot, with the same slope as the first eigenvector of the 2x2 covariance matrix of the 2xN time series of pairs. The first eigenvector points in the direction of maximum variance, thus minimizing the residuals perpendicular to the line, rather than in an arbitrarily chosen y-direction.

**3.1.1 Bivariate regression**

We thank the reviewer to make us aware of the problems with the ordinary least squares regression that seems to be largely unknown in the atmospheric sciences. The reviewer asked in particular to compute a bivariate regression fit for the bivariate  $T_{12}$  distribution shown in figure 2 and provided a recipe for the procedure. We followed the suggestion and got the following results: The covariance matrix for the original intercalibrated data pairs is

$$C = \left(\begin{array}{rrr} 23.0041 & 19.0753\\ 19.0753 & 22.7789 \end{array}\right)$$

The eigenvalues are 41.9671 and 3.81587, reflecting that the data cloud is much more elongated along the diagonal than perpendicular to it. The first eigenvector is proportional to  $(1, 0.994114)^T$ , that is, the slope of the regression is very close to unity when the errors in the data on the x-axis are taken into account. (Of course, the second eigenvector is perpendicular to the first). According to York et al. (2004, eq. 13a), the regression line crosses the bivariate mean, that is in our case (240.029, 240.663) and thus the intercept of the bivariate regression line is 2.04681. The bivariate regression coefficients indicate that indeed, a bivariate least squares fit as suggested by the reviewer fits better to the data than the ordinary least squares fit. These results show again how good the original intercalibration by Shi and Bates (2011) was.

Figure 2 shows the new figure 2 in the revised manuscript, displaying again the ideal fit line (the diagonal y = x, dashed), and the ordinary least squares fit (solid). Additionally we have plotted the bivariate least squares fit (dashed-dotted) that represents the best fit when the errors in the independent values (here  $T_{12}$  (N15)) are taken into account.

Note that the marginal means of  $T_{12}(N14)$  follow rather the ordinary least squares than the bivariate regression line (shown in the new figures 2 and 6 in the revised paper). Thus a regression of the 1st kind (which uses the marginal means for correction) is closer to the traditional regression line than to a bivariate regression line.

**3.1.2 Using bivariate regression for correction is inconsistent**

Finally we want to point out that while the bivariate regression line provides in some sense the best fit through a bivariate distribution of data with uncertainties in both dimensions, it seems just therefore inappropriate to derive from it corrections to the quantity on the x-axis. To correct x needs a fixed value of x, as the OLS regression and regression of the first kind assume. If, however, uncertainties in the x-dimension of the data are explicitly considered, it is not immediately clear to which value the correction should be applied or how it may be derived and formulated.

Look at the sketch given in fig. 3 of this reply. Regression works on data pairs  $(x_i, y_i)$  and determines from these a best straight line that minimises a certain distance norm. In the case of OLS regression the sum of the lengths of the solid red lines is minimised and the result is a straight line a + bx. In the case of bivariate regression the sum of the lengths of the solid blue lines is minimised, resulting in a straight line A + Bx. In the sketch we have plotted only one fit line for convenience, but usually |B| > |b|. Note that the OLS regression yields the prediction  $y_{OLS}$  as a function of  $x_i$  only, once the fit coefficients are given, while the bivariate regression predicts  $y_a$  as a function of  $x_i$  and  $y_i$  (once the coefficients A, B are known). Independent data which are to be corrected using the regression fits do not come as pairs, there is only data  $x_o$ . An OLS-type correction is possible here as  $x_{o,corr} = a + bx_o$ . Of course, a similar correction can be performed using the coefficients (A, B) of the bivariate fit. However, this correction is inconsistent with the bivariate regression, which should properly involve  $x_{o,a}$ , the adjusted x-coordinate that belongs to the actually observed value,  $x_o$ . However,  $x_{o,a}$  cannot be determined, since a corresponding datum  $y_o$  is not given.

Thus, in regression-type correction one has two non–ideal possibilities:

- a) to use the non-ideal OLS fit and make the corrections consistently
- b) to use the ideal bivariate fit and make the corrections inconsistently.

To illustrate these arguments we apply the correction here to the "training" data, that is, the data pairs from which the regression coefficients have been determined.

The correction that is consistent with the bivariate regression is in our case:

$$\hat{T}_{12/15} = A + \frac{B}{1+B^2} \left( T_{12/15} + B T_{12/14} - AB \right).$$
(1)

(The formula can be derived from the equations given by York et al., 2004, assuming independent data and equal weights). It contains  $T_{12/14}$  explicitly and this shows that the formula cannot be applied for correction of independent data.

The inconsistent correction is simply

$$\hat{T}_{12/15} = A + B \, T_{12/15}. \tag{2}$$

The result of both excercises is plotted in fig. 4 of this reply. Inconsistent use of the regression coefficients leads to the blue points that are arranged on the bivariate fit line. They are all above the black diagonal, that is, the brightness temperatures are corrected upward as they should, at least in the lower tail of the data. The correction above 240 K is not really necessary as we have demonstrated in the manuscript. The red dots represent the result of consistent use of the regression coefficients. As desired, in the low tail this leads to an upward correction. But at higher temperatures the corrections go up and down, perhaps with an average correction close to zero. But this leads to noise that is unwanted, and — as we have seen — it is unnecessary. According to these findings, we cannot derive corrections for satellites beyond N15 using the consistent formula (1), simply because there are no  $T_{12/14}$  data beyond 2005. Corrections using the inconsistent formula (2) would not be appropriate to be derived.

**3.2 Influence of close data pairs**

I think the authors should be more clear how they choose pairs of data points. For example 2 HIRS/2 points "A" and "B", and 2 HIRS/3 points "1" and "2", if all close together, can produce 4 pairs for comparison: A-vs-1, A-vs-2, B-vs-1, and B-vs-2. Do the authors avoid this sort of inflation?

We do not construct data pairs from data in adjacent grid points. A data pair consists of two daily averages, one from N14 measurements and one from N15 measurements, in the same grid box. We have added a few sentences in section 2.1 and hope that the new text is clear now.

**3.3** Minor corrections and comments**

**3.3.1 Figures and colour tables**

We have remade the scatter plots with slightly larger symbols (for the expert: In Gnuplot we use "points" now with pointsize 0.3 instead of "dots"). The rainbow colour scheme has been replaced by a blue-yellow-red colour series which seems to be distinguishable by colour-blind readers. Without knowing what is meant with "perceptually uniform", we hope that the new colour table meets that category as well.

**3.3.2 Others**

P1 7: Ok

P2 1-3: ok, references added.

P3 27-28: ok, in section 4. Added.

- P4 29-30: This was meant as a mathematical bracket because both numbers have the unit K. We have removed all brackets in the revised version.
- P7 26-27: We have modified the sentence into "Although it looks like a manifestation of climate change it is rather a manifestation of the change from HIRS 2 to HIRS 3".
- Fig. 9: yes, that was wrong. Corrected.

**References**

Please look in the revised manuscript's reference section for the literature quoted in this reply.